# Oral Fluid Biomarkers for Diagnosing Gingivitis in Human: A Cross-Sectional Study

**DOI:** 10.3390/jcm9061720

**Published:** 2020-06-03

**Authors:** Inpyo Hong, Hyung-Chul Pae, Young Woo Song, Jae-Kook Cha, Jung-Seok Lee, Jeong-Won Paik, Seong-Ho Choi

**Affiliations:** Department of Periodontology, Research Institute of Periodontal Regeneration, Yonsei University College of Dentistry, Seoul 03722, Korea; inpyo@yuhs.ac (I.H.); samdasoo10@yuhs.ac (H.-C.P.); tigger09@naver.com (Y.W.S.); chajaekook@gmail.com (J.-K.C.); cooldds@gmail.com (J.-S.L.); jeongwon100@hotmail.com (J.-W.P.)

**Keywords:** gingivitis, matrix metalloproteinase-8 (MMP-8), myeloperoxidase (MPO), biomarker, diagnosis

## Abstract

Diagnoses based on oral fluid biomarkers have been introduced to overcome limitations of periodontal probe-based diagnoses. Diagnostic ability of certain biomarkers for periodontitis have been identified and widely studied, however, such studies targeting gingivitis is scarce. The aims of this study were to determine and compare the efficacies and accuracies of eight biomarkers in diagnosing gingivitis with the aid of receiver operating characteristic (ROC) curves. The probing depth (PD), clinical attachment loss (CAL), bleeding on probing (BOP), gingival index (GI), and plaque index (PI) were examined in 100 participants. Gingival crevicular fluid was collected using paper points, and whole-saliva samples were collected using cotton roll. Samples were analyzed using enzyme-linked immunosorbent assay kits for the different biomarkers. The levels of matrix metalloproteinase (MMP)-8, MMP-9, lactoferrin, cystatin C, myeloperoxidase (MPO), platelet-activating factor, cathepsin B, and pyridinoline cross-linked carboxyterminal telopeptide of type I collagen were analyzed. MPO and MMP-8 levels in saliva were strongly correlated with gingivitis, with Pearson’s correlation coefficients of 0.399 and 0.217, respectively. The area under the curve (AUC) was largest for MMP-8, at 0.814, followed by values of 0.793 and 0.777 for MPO and MMP-9, respectively. The clinical parameters of GI and PI showed strong correlations and large AUC values, whereas PD and CAL did not. MMP-8 and MPO were found to be effective for diagnosing gingivitis. Further investigations based on the results of this study may identify clinically useful biomarkers for the accurate and early detection of gingivitis.

## 1. Introduction

Periodontitis is a complex inflammatory reaction against a dysbiotic challenge that results in the irreversible loss of periodontal supporting tissues. The high global prevalence of periodontitis results in a high burden in terms of productivity loss and the costs of treating oral disease [1,2]. Since gingivitis is considered a prerequisite to periodontitis, early intervention through the accurate and early diagnosis of the gingivitis state is an effective approach for preventing periodontitis [3].

Periodontal disease is conventionally diagnosed based on measurements of clinical attachment loss (CAL) and bleeding on probing (BOP) using a periodontal probe. However, periodontal probing involves pain to patients and has standardization limitations due to variations of the insertion pressure and angle between clinicians [4,5]. Moreover, CAL can only be measured after the significant breakdown of more than 2 mm of periodontal tissue has occurred, and it only shows the history of disease progression rather than demonstrating the present inflammatory state of the disease [6].

The use of biomarkers in oral fluids such as saliva and gingival crevicular fluid (GCF) has been suggested for overcoming the limitations of probing-based diagnoses of periodontal disease [7,8,9]. Cytokines and proteins involved in host responses are potentially useful biomarkers as they directly reflect the current inflammatory state of the periodontal tissue [10]. Moreover, saliva and GCF are easy to collect from patients and contain various biomarkers, which has resulted in numerous studies suggesting the use of various oral fluid biomarkers for diagnosing periodontal disease.

Previous studies have suggested the following possible biomarkers for periodontal disease: matrix metalloproteinase (MMP)-8, MMP-9, lactoferrin, cystatin C, myeloperoxidase (MPO), platelet-activating factor (PAF), cathepsin B, and pyridinoline cross-linked carboxyterminal telopeptide of type I collagen (ICTP) [11,12,13,14,15]. MMP-8, MMP-9, MPO, and cathepsin B are tissue destructive enzymes that are activated in inflammatory cascade, and ICTP is a bone degradation marker. Lactoferrin, cystatin C and PAF are known as markers for chronic inflammation. Thus, it is obvious that these biomarkers are pathophysiologically related to periodontal inflammation. However, while the level of each of these biomarkers differed significantly between disease and normal groups in previous studies, such significant differences do not directly indicate that the biomarkers have clinically useful diagnostic power.

Receiver operating characteristic (ROC) curves have been used in medicine to identify the effectiveness and power of biomarkers for the differential diagnosis of disease. ROC curves plot the sensitivity versus the specificity of each biomarker, and they can be used to compare the diagnostic capabilities of biomarkers based on the area under the curve (AUC). These curves are especially effective in comparing the diagnostic capabilities of biomarkers on different units [16].

Previous studies have demonstrated that MMP-8, MMP-9, lactoferrin, cystatin C, MPO, PAF, cathepsin B, and ICTP can be used to diagnose periodontitis. However, to the best of our knowledge, few studies have investigated the correlations between these biomarkers and gingivitis. In addition to early detection of gingivitis, confirming biomarkers of gingivitis can be the basis for future studies regarding gingival health state and individual variation of disease susceptibility. Thus, the aims of this study were to: (1) determine the efficacies and accuracies of eight biomarkers in diagnosing gingivitis, and (2) compare the diagnostic abilities of the biomarkers using ROC curves.

## 2. Materials and Methods

### 2.1. Study Design and Population

One hundred and six voluntary participants were recruited at the clinic of the Department of Periodontology, Yonsei University Dental Hospital. The study was conducted in accordance with the Declaration of Helsinki and approved by the Institutional Review Board of Yonsei University Dental Hospital (Approval number: 2-2016-0044). Each voluntary participant was given verbal and written information about the study and then signed an informed consent form before being enrolled in the study. The study design is illustrated in Figure 1. Before enrollment, all volunteers were clinically examined and interviewed in accordance with the inclusion and exclusion criteria listed below.

### 2.2. Inclusion and Exclusion Criteria

The inclusion criteria were: (1) being >18 and <80 years of age and in good general health, (2) having a minimum of 18 teeth, (3) having less than 5 mm of CAL at the first visit, and (4) being diagnosed as chronic gingivitis or periodontal health according to the consensus report of the 2017 world workshop [17]. The exclusion criteria were: (1) not providing written informed consent, (2) being pregnant or lactating, (3) having a severe systemic disease such as uncontrolled diabetes or hypertension, (4) having enrolled in another clinical study within the previous three months, (5) taking antiplatelet agents or anticoagulants or having a history of hemorrhage or disease, (6) taking an antibiotic within the previous month, (7) having an oral mucosal inflammatory condition (e.g., lichen planus or leukoplakia), or (8) judged as being inappropriate for study inclusion for some other reason by the clinician.

### 2.3. Clinical Evaluation

A clinician in the clinic at the Department of Periodontology measured periodontal parameters in the following six representative teeth for each subject: #16, #21, #24, #36, #41, and #44. The probing depth (PD), BOP, and CAL were measured at mesiobuccal, midbuccal, distobuccal, mesiolingual, midlingual, and distolingual sites for each tooth. The gingival index (GI) and plaque index (PI) were also measured for each tooth based on Turesky et al. [18].

In accordance with the consensus report of the 2017 World Workshop, patients exhibiting BOP at more than 10% of the investigated sites were diagnosed as gingivitis [17]. Since six sites of the representative six teeth were examined for BOP in this study, patients who had more than four BOP sites were classified into the gingivitis group.

### 2.4. GCF and Saliva Collection

Methods for collecting GCF and saliva were set up based on previous studies [9,19]. Patients were asked to fast at least 8 h before visiting the clinic. The site at which GCF was collected was dried with an air syringe and isolated from salivary and blood contamination with the aid of cotton rolls or mirror retraction. Supragingival plaques were carefully removed using gauze and paper points (Dia-Pro ISO.04, DiaDent, Cheongju, Korea) that were gently inserted into the gingival crevice mesial, midbuccal, and distal of representative three teeth (#16, #24, and #36). The paper points were removed after 30 s and then stored together in a centrifuge tube, into which 300 μl of phosphate-buffered saline was added before being incubated at 4 °C overnight. After centrifugation at 3000× *g* for 5 min at 4 °C, the clear supernatant was extracted and stored at –80 °C until being analyzed.

Patients rinsed their mouths with pure water prior to saliva collection. Whole-saliva samples were then collected by keeping cotton rolls in the mouth for 60 s, which were then stored in a centrifuge tube. After centrifugation at 3000× *g* for 10 min, a clear saliva sample yielded in a conical tube was stored at –80 °C until being analyzed.

### 2.5. Enzyme-Linked Immunosorbent Assay Analysis of Molecular Biomarkers

Enzyme-linked immunosorbent assay (ELISA) was conducted based on previous studies and manufacturers’ methods [11,13,14,15]. The PAF level in GCF was measured using an enzyme-linked immunosorbent assay (ELISA) kit (Biomatik, Cambridge, ON, Canada), as were the cathepsin B and ICTP levels (Aviva Systems Biology, San Diego, CA, USA). The analysis was performed according to the manufacturers’ methods. Each sample was added to a microplate well that had been precoated with affinity polyclonal antibodies specific for PAF, cathepsin B, and ICTP. After washing, specific enzyme-conjugated polyclonal antibodies and substrate solutions were added to the wells. After confirming the color change in the reaction (15–30 min), the stop solution was added and the absorbance was measured at 450 nm in a microplate reader (Infinite M200 PRO NanoQuant microplate reader, TECAN, Zurich, Switzerland). The sample values were calculated using a standard curve and the levels of molecular biomarkers were expressed in picograms per milliliter.

The cystatin C, MPO, and MMP-9 levels in saliva were measured using an ELISA kit (R&D Systems, Minneapolis, MN, USA), as were the lactoferrin and MMP-8 levels in saliva (Biovendor, Brno, Czech Republic). The analysis was performed according to the manufacturers’ methods. For each assay, the absorbance was measured at 450 nm and calculated using a standard curve, with the levels of molecular biomarkers expressed in nanograms per milliliter.

### 2.6. Statistical Analysis

Commercially available statistical analysis software (IBM SPSS Statistics 23, SPSS, Chicago, IL, USA) was used to perform the statistical analyses. The Mann–Whitney U test was used to compare demographic and clinical parameters between the gingivitis and healthy groups. Pearson’s correlation analysis was used to identify correlations between the biomarker levels and the percentage of BOP sites. The correlations between clinical parameters and the percentage of BOP sites were also analyzed.

The diagnostic ability of each marker was evaluated by constructing a ROC curve, from which the AUC was calculated. The sensitivity and specificity were calculated for each point on the ROC curve. The cutoff for each biomarker was defined as the value that was farthest from the reference line in the ROC curve.

## 3. Results

### 3.1. Demographic Analysis

The demographic and clinical parameters of the participants are summarized in Table 1. Among 100 subjects, 85 were diagnosed as gingivitis, and 15 were diagnosed as gingival health. Age, sex, and CAL did not differ significantly between the healthy and gingivitis groups. As expected, BOP, PI, and GI were significantly larger in the gingivitis group, while there was no intergroup difference in CAL.

### 3.2. Correlation Analyses

The coefficients of the biomarkers and clinical parameters are summarized in Table 2. Since, percentage of BOP sites is the only criterion for clinical diagnosis of gingivitis, correlations of each biomarkers and percentage of BOP sites were analyzed [17]. MPO was the biomarker that showed the strongest correlation with the percentage of BOP sites, with a Pearson’s correlation coefficient of 0.399 (*p* < 0.01). MMP-8 showed a positive correlation with the percentage of BOP sites, with a Pearson’s correlation coefficient of 0.217 (*p* < 0.05). The other analyzed biomarkers did not show significant correlations with the percentage of BOP sites (*p* > 0.05).

The clinical parameters of GI and PI showed strong correlations with the percentage of BOP sites (*p* < 0.01), while CAL did not show any correlation with the percentage of BOP sites.

### 3.3. ROC Curves

ROC curves of the biomarkers and clinical parameters for gingivitis are shown in Figure 2, and the corresponding AUC values are summarized in Table 3. MMP-8 was the biomarker that showed the largest AUC of 0.734, with a cutoff of 6.46 ng/mL. MMP-9 and MPO also showed large AUC values, of 0.703 and 0.685, respectively, while cystatin C showed an AUC of 0.667. The ROC curves for MMP-8, MMP-9, MPO, and cystatin C are shown in Figure 3. The AUC values for lactoferrin, PAF, ICTP, and cathepsin B levels were lower than 0.5, indicating that these parameters were not statistically relevant.

The AUC values of the clinical parameters are summarized in Table 4. GI and PI showed large AUC values of 0.788 and 0.692, respectively, while that of CAL was 0.532, indicating that this was not suitable for diagnosing gingivitis.

## 4. Discussion

This study demonstrated that the MMP-8 and MPO levels are suitable for diagnosing gingivitis based on significant results obtained in both correlation and ROC curve analyses. In contrast to most previous studies focusing on differences in the concentrations of biomarkers between periodontitis and healthy groups, our study evaluated and compared the diagnostic powers of different biomarkers for gingivitis [11,13,15,20]. Few previous studies have focused on diagnosing gingivitis. One study included an experimental gingivitis group, but the evaluation of the diagnostic abilities of biomarkers was limited to the periodontitis group only [21].

Gingivitis has been defined as “an inflammatory lesion which remains confined to the gingiva and does not extend to the periodontal attachment (cementum, periodontal ligament and alveolar bone). Such inflammation is reversible by reducing levels of dental plaque at and apical to the gingival margin” [17]. As this definition implies, gingivitis itself is not a state of tissue destruction and is reversible to a healthy state. Gingivitis usually shows clinical features such as redness and swelling, but it is commonly painless, which means that most affected patients are not aware of it [22]. However, since untreated gingivitis may progress to periodontitis accompanied by irreversible tissue destruction, it is important to diagnose periodontal disease at the gingivitis stage before the patient experiences clear symptoms or the loss of periodontal attachment occurs [3]. It is therefore desirable to identify biomarkers for the early detection of gingivitis before it progresses to periodontitis.

Previous studies have found that the concentrations of numerous oral fluid biomarkers differ between normal and periodontitis groups [10,11,13,14,23]. A statistically significant difference in concentrations or a strong correlation shows the possibility of applying such biomarkers to disease diagnosis. However, the presence of such a concentration difference between healthy and diseased groups does not imply that a particular biomarker has a strong diagnostic ability [24]. Instead, the clinical utility of a biomarker in diagnoses is related to its ability to discriminate between patients who have and those who do not have a disease according to criteria based on the concentration of the biomarker. Thus, the pure existence of a difference between healthy and diseased groups does not fulfill the conditions of a biomarker being appropriate for diagnoses, since both clinical relevance and validity are also needed [25].

An ROC curve is a plot of sensitivity versus 1 minus specificity, and it is one of the most popular and effective methods for evaluating the discrimination ability of a diagnosis model [26]. In detail, an ROC curve can provide evidence for optimal cutoff values for a dichotomous diagnosis, and the corresponding AUC is considered an effective parameter for comparing the accuracies of different diagnostic models [27]. The construction of a ROC curve involves plotting multiple points rather than making a single measurement, which provides the advantage of the AUC being independent of any particular reference value used in a diagnosis and so is suitable for evaluating and comparing utility of different biomarkers in different units.

MMP-8 showed the largest AUC and strongest correlation with gingivitis in this study. MMP-8 is one of the most widely studied host-derived enzymes as a biomarker for periodontal disease, which is related to tissue destruction [9,15,28]. In a state of periodontal disease, neutrophil and macrophages release MMPs as part of the host response [20]. Among MMPs, MMP-8 degrades collagen types 1 and 3, which are major components of periodontal tissues [28]. Previous studies have found that individual patients diagnosed with periodontitis showed significant elevations of the MMP-8 concentration compared to healthy groups [12,15]. Also, MMP-8 can be used to predict the progression of periodontal disease or the response to periodontal therapy [19,29].

MPO also showed a large AUC and a strong correlation with gingivitis in the present study. A previous study observed a difference in MPO concentration between healthy and chronic periodontitis group [13]. MPO is expressed from phagosome of neutrophils and reacts with hydroxyl peroxide to form metabolites that play a significant role in bactericidal activity [30]. Since MPO mediates MMP-8 activation and collagenolytic MMP activation cascade, MPO produced in initial immune response of neutrophil amplifies the tissue destructive process [31]. Conversely, elevated concentration of MPO itself can reinforce pathogenic challenge or gingival inflammation. A recent study suggested that high a concentration of MPO can contribute to dysbiosis of the gingival sulcus microbiome [32].

Leppilahti et al. found that the MMP-8 and MPO levels in GCF showed large AUC values for the diagnosis of chronic periodontitis, of 0.90 and 0.98, respectively, with cutoff values of 2520.6 ng/mL and 135.9 ng/mL, which produced sensitivities and specificities of around 0.9 [13]. MMP-8, MMP-9, and MPO showed smaller AUC values and lower cutoff values in the present study. Also, the diagnostic sensitivity was similar, but the diagnostic specificity was significant lower in our study than in that of Leppilahti et al. These differences in diagnostic power are due to the differences in the targeted diseases: the diagnosis target in the previous study was periodontitis, while that in our study was gingivitis, which is a mild and early state of periodontitis. Differences in the severity of periodontal inflammation result in differences in the variety and concentrations of inflammatory substances [33]. Thus, gingivitis shows lower concentrations of inflammatory biomarkers compared to periodontitis, and diagnostic biomarkers are more difficult to identify when their concentrations are lower in the diseased state due to the inevitable noise in tests affecting the results more when the reference concentration is low. Also, performing diagnoses with a low reference concentration is vulnerable to external confounding factors such as fluctuations in the general condition and contamination of the sample fluid. The consequence of all of these factors make that it is difficult for a diagnostic model with a low reference concentration to be clear and powerful, as represented by a large AUC.

An ideal diagnostic model would have a high sensitivity and specificity simultaneously, but a diagnostic model with a high sensitivity and low specificity can be a better alternative to one with a low sensitivity and high specificity [34]. In the case of the gingivitis, in which early intervention is especially important, and treatment or a diagnostic test is not invasive or risky to patients, a false negative is more problematic than a false positive. Also, treating gingivitis using methods such as scaling as part of maintenance therapy does not have harmful effects on patients.

Clarifying the biomarkers of gingivitis can be the basis for future studies that clearly identify the effect of dietary intake of micronutrients on periodontal health. It is widely accepted that micronutrient supplements, such as vitamins, can induce alterations of the inflammatory cascade and reduce chronic inflammation as a result [35]. Supporting studies evaluated periodontal health or inflammation by probing-based conventional clinical methods or changes of certain oral fluid biomarkers [36,37].

However, in a probing-based evaluation, there is limitation of standardization due to intra- and inter-examiner variation [38]. Also, in case of mild inflammation such as gingivitis, it is difficult to discriminate BOP due to actual inflammatory and BOP due to trauma of periodontal probing which is false positive [6,39]. In biomarker-based evaluation, previously documented oral fluid biomarkers showed a difference in concentration between periodontitis diagnosed patients and healthy individual. Since gingivitis is prerequisite to periodontitis, it seems to be natural to regard that both gingivitis and periodontitis can share diagnostic biomarkers. However, there is difference in severity and stage of inflammation, therefore it could be impractical to regard all diagnostic biomarkers for periodontitis to have proper diagnostic ability in gingivitis. To the best of our knowledge, there was not any study that aims to confirm biomarkers for gingivitis with a large sample size. The sample size of this study was sufficient, thus the result of this study can contribute to clarifying biomarkers of gingivitis.

In summary, MMP-8 and MPO have now been shown to be effective in diagnosing gingivitis. Nevertheless, since gingivitis is a milder and earlier inflammatory state compared to periodontitis, a biomarker-based diagnosis method for gingivitis needs to be more sensitive. Further investigations based on the findings of this study may lead to clinical diagnosis methods based on biomarkers that facilitate the accurate and early detection

## Figures and Tables

**Figure 1 jcm-09-01720-f001:**
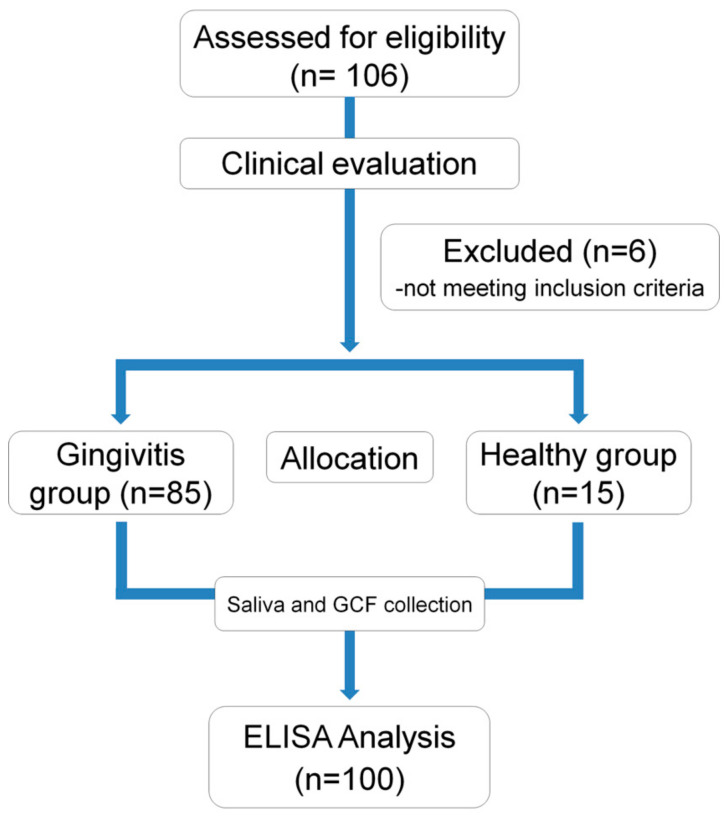
Flowchart and study protocol. GCF, gingival crevicular fluid; ELISA, enzyme-linked immunosorbent assay.

**Figure 2 jcm-09-01720-f002:**
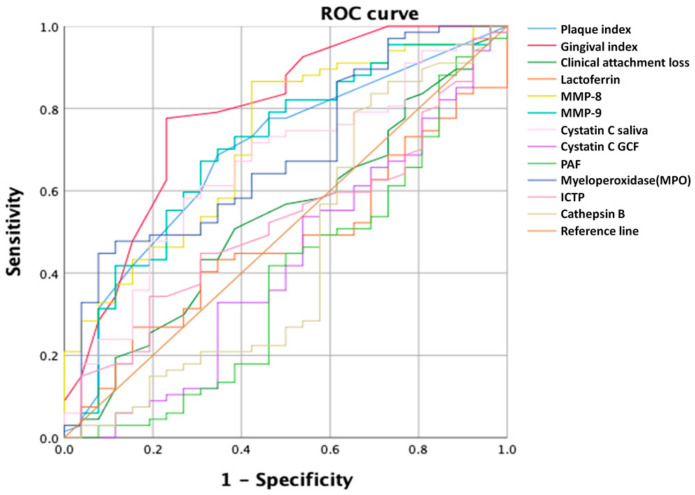
Receiver operating characteristic (ROC) curve of clinical parameters and biomarkers. Only plaque index, gingival index, MMP-8, MMP-9, MPO, and cystatin C showed convex shape of curve. Concave curve means subject of ROC curve does not have any diagnostic ability. As the diagnostic power of subject is increased, convexity of ROC curve increases. MPO, myeloperoxidase; MMP-8, matrix metalloproteinase-8; MMP-9, matrix metalloproteinase-9; GCF, gingival crevicular fluid; PAF, platelet activating factor; ICTP, pyridinoline cross-linked carboxyterminal telopeptide of type I collagen.

**Figure 3 jcm-09-01720-f003:**
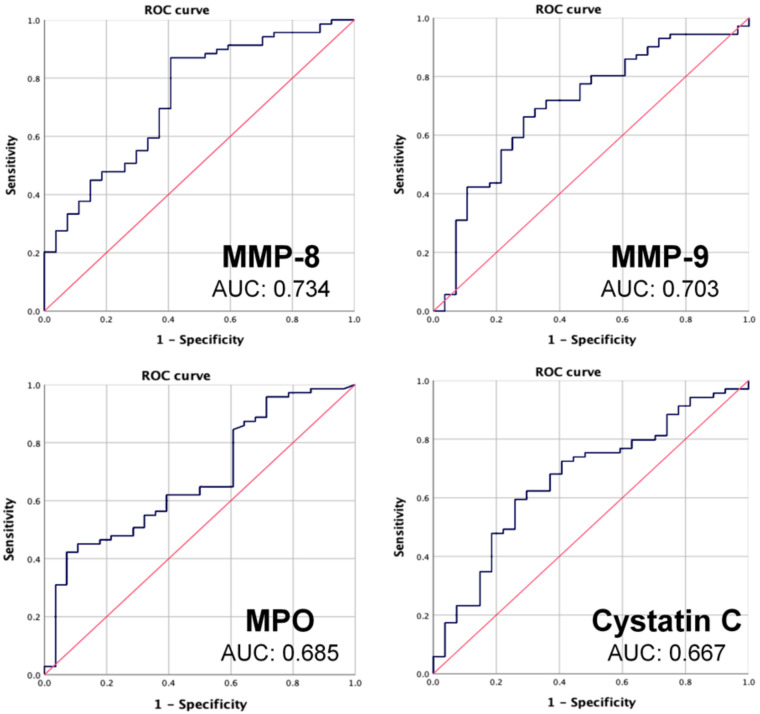
Receiver operating characteristic (ROC) curve of MMP-8, MMP-9, MPO, and cystatin C. Area under ROC curve (AUC) is measured based on the curve. The farthest point from standard line is selected to set cut-off value for diagnosis. MPO, myeloperoxidase; MMP-8, matrix metalloproteinase-8; MMP-9, matrix metalloproteinase-9.

**Table 1 jcm-09-01720-t001:** Distribution of study subject characteristics (mean ± standard deviation) in healthy and gingivitis group.

	Healthy Group (*n* = 15)	Gingivitis Group (*n* = 85)	*p*-Value
Age (years)	34.93 ± 15.79	32.65 ± 12.21	0.85
Female/Male (n)	7/8	47/38	0.54
BOP site (%)	5.56 ± 0.89	26.96 ± 4.70 *	<0.001
CAL (mm)	2.55 ± 0.30	2.60 ± 0.27	0.71
PI	0.13 ± 0.22	0.53 ± 0.39 *	<0.001
GI	0.39 ± 0.35	0.96 ± 0.37 *	<0.001

^*^ Statistically significant difference compared to the healthy group (*p* < 0.05). BOP, bleeding on probing; CAL, clinical attachment loss; PI, plaque index; GI, gingival index.

**Table 2 jcm-09-01720-t002:** Correlation of clinical parameter and biomarkers with number of BOP sites.

	Coefficient of Correlation	*p*-Value
Average plaque index (PI)	0.411 **	<0.001
Average gingival index (GI)	0.766 **	<0.001
Clinical attachment loss (CAL)	0.096	0.346
MPO	0.399 **	<0.001
MMP-8	0.217 *	0.034
MMP-9	0.032	0.750
Cystatin C	0.055	0.595
Lactoferrine	0.137	0.183
PAF	0.034	0.742
Cathepsin B	-0.093	0.366
ICTP	-0.004	0.966

^*^ Statistically significant correlation (*p* < 0.05). ** Statistically highly significant correlation (*p* < 0.01). BOP, bleeding on probing; MPO, myeloperoxidase; MMP-8, matrix metalloproteinase-8; MMP-9, matrix metalloproteinase-9; PAF, platelet activating factor; ICTP, pyridinoline cross-linked carboxyterminal telopeptide of type I collagen; BOP, bleeding on probing; CAL, clinical attachment loss; PI, plaque index; GI, gingival index.

**Table 3 jcm-09-01720-t003:** Receiver operating characteristic (ROC) area under curve (AUC) of biomarkers and clinical parameters. Sensitivity and specificity of each biomarker were selected at the cut-off value points of the ROC curve.

	AUC	Cut-Off Value (ng/mL)	Sensitivity	Specificity
MMP-8	0.734	6.464	0.87	0.60
MMP-9	0.703	38.075	0.739	0.63
MPO	0.685	12.75	0.87	0.42
Cystatin C	0.667	59.28	0.725	0.59

MPO, myeloperoxidase; MMP-8, matrix metalloproteinase-8; MMP-9; matrix metalloproteinase-9.

**Table 4 jcm-09-01720-t004:** Area under ROC curve (AUC) of clinical parameters.

	AUC
Gingival index	0.788
Plaque index	0.692
Clinical attachment loss	0.532

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
