# Peer review of "Oral Fluid Biomarkers for Diagnosing Gingivitis in Human: A Cross-Sectional Study"

_jcm, 2020, doi:10.3390/jcm9061720_

Round 1

Reviewer 1 Report

The study provides interesting information which will require ratification in subsequent studies.

The meaning of CAL must be put in the abstract

In my opinion, the main weakness of the study is that the sample size had not been previously calculated, moreover the authors have not explained how the sample was selected.

Author Response

The study provides interesting information which will require ratification in subsequent studies.

The meaning of CAL must be put in the abstract

Response 1: Thank you for your comment. We put the meaning of CAL in the abstract.

“CAL -> clinical attachment loss (CAL)”

In my opinion, the main weakness of the study is that the sample size had not been previously calculated, moreover the authors have not explained how the sample was selected.

Response 2: Thank you for your accurate comment. This study was cross-sectional study to find out which biomarker could be the most effective, not to find out prevalence of the disease, thus, it was thought to be inappropriate to select a particular preliminary study to facilitate sample size calculation. Therefore, we referred previous study (Leppilathi et al. 2014) which identified diagnostic accuracy of gingival crevicular fluid biomarker. In Leppilathi et al. there were 39 subjects (Healthy=20, Gingivitis=19) and, within this sample size of 39, significant diagnostic ability of myeloperoxidase and matrix metalloproteinase-8 in gingivitis were found. So, we thought the sample size of 100 could be enough to discriminate diagnostic ability of oral fluid biomarkers.

Volunteered participants were given verbal and written information about the study and after the consents, they were selected as the sample of this clinical study. As your advice, we changed our manuscript, “participants” to “voluntary participants”.

We truly appreciate to your comment. Thank you.

Reviewer 2 Report

More precision of  inclusion criteria is required 

Why have authors choses chosen the age span of 18-80 yrs old?

Although the control and disease group are somehow matched, The age spectrum is  wide. It is not clear how the physiologic effect of age on the state of gingiva homeostasis and its correlation with clinical parameter is addressed ? 

Author Response

More precision of inclusion criteria is required

Response 1: Thank you for your comment. Inclusion criteria that “being diagnosed as chronic gingivitis or periodontal health” implies more precise information. In 2017 world workshop, gingivitis case was defined as patients exhibiting BOP at more than 10% of the investigated sites. In clinical evaluation part, there is precise information about gingivitis case definition.

As your advice, we added inclusion criteria “(4) being diagnosed as chronic gingivitis or periodontal health according to the consensus report of the 2017 World Workshop.

Reference) Chapple, I.L.C.; Mealey, B.L.; Van Dyke, T.E.; Bartold, P.M.; Dommisch, H.; Eickholz, P.; Geisinger, M.L.; Genco, R.J.; Glogauer, M.; Goldstein, M., et al. Periodontal health and gingival diseases and conditions on an intact and a reduced periodontium: Consensus report of workgroup 1 of the 2017 World Workshop on the Classification of Periodontal and Peri-Implant Diseases and Conditions. J Periodontol 2018, 89 Suppl 1, S74-S84,

Why have authors choses chosen the age span of 18-80 yrs old?

Response 2: Since there were studies using biomarkers to diagnose gingivitis and periodontal disease, we referred criteria of previous studies.

Ref 1) Nascimento et al. Salivary levels of MPO, MMP-8 and TIMP-1 are associated with gingival inflammation response patterns during experimental gingivitis. Cytokine. 2019:115:135-141

Ref 2) Lee C-H et al. The potential of salivary biomarkers for predicting the sensitivity and monitoring the response to nonsurgical periodontal therapy: A preliminary assessment. J Periodont Res. 2018;53:545‐554

Ref 3) Hong et al. A randomized, double-blind, placebo controlled multicenter study for evaluating the effects of fixed-dose combinations of vitamin C, vitamin E, lysozyme, and carbazochrome on gingival inflammation in chronic periodontitis patients. BMC Oral Health. 2019:19:40

Ref 4) Ito et al. Relationship between antimicrobial protein levels in whole saliva and periodontitis. J Periodontol. 2008:19:2

Although the control and disease group are somehow matched, The age spectrum is wide. It is not clear how the physiologic effect of age on the state of gingiva homeostasis and its correlation with clinical parameter is addressed?

Response 3: We truly agree to your comment. We also concede that physiologic effect of aging could related to individual susceptibility or resistance to microbial challenge which induce gingival inflammation. So wide age spectrum of subject could interfere the result of study. However, as reviewer noticed, there was no significant difference of age in control and disease group. Thus, similarity of age distribution could negate its effect on diagnosis. Also, age per se is not a significant risk factor in periodontitis (Van Dyke. 2005). Lack of oral hygienic management and increased prevalence of systemic disease due to aging could be main risk of aging to gingival health (Abdellatif et al. 1987). However, inclusion criteria which only include gingivitis case, naturally exclude subjects with such conditions, thus we thought it would be obscure to consider effect of age in interpreting results of this study. In future related study, we fully agree that we should restrict subject age spectrum or analyze the subject separately according to age, following your comment.

Ref) Van Dyke. T et al. Risk factors for periodontitis. J Int Acad Periodontol. 2005:7(1), 3-7.

Ref) Abdellatif, H. M., & Burt, B. A. An Epidemiological Investigation into the Relative Importance of Age and Oral Hygiene Status as Determinants of Periodontitis. Journal of Dental Research, 1987: 66(1), 13–18.

We truly appreciate to your accurate comments. Thank you.